# The Extended Merge Hypothesis and the Fundamental Principle of Grammar

**Norbert Hornstein**

Department of Linguistics, University of Maryland, College Park, MD 20742, USA; NHORNSTE@UMD.EDU

**Abstract:** This paper discusses the main minimalist theory within the Minimalist Program, something I dub the (Weak) Merge Hypothesis (MH). (1) The (Weak) Merge Hypothesis (MH): Merge is a central G operation. I suggest that we extend (1) by adding to it a general principle that I dub the Fundamental Principle of Grammar (FPG). (2) The Fundamental Principle of Grammar (FPG): α and β can be grammatically related. (G-related) only if α and β have merged. Adding (1) and (2) gives us the Strong Merge Hypothesis. (3) The Strong Merge Hypothesis (SMH): All grammatical dependencies are mediated by Merge. SMH has some interesting consequences which the rest of the paper briefly reviews. Highlights include the Movement Theory of Construal, The Periscope Property on selection, as well as preserving the standard results from the Weak Merge Hypothesis.

**Keywords:** extended merge hypothesis; fundamental principle of grammar; merge

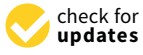



0. As I see it, the Minimalist Program (MP) is the third stage in the generative investigation of the properties of the faculty of language (FL) (See Hornstein, forthcoming, 2018 and 2019 for some discussion [1–3]). The central project of the first stage was the description of the grammatical capacities of native speakers of a particular language. The second stage took as cynosure the description of the species-specific meta-capacity that allows humans to become native speakers of a particular language. MP is the next logical step in this inquiry and it takes as its central project to explain why the meta-capacity humans have to acquire language specific grammars (G) has the specific properties it has. The first two projects (NOTE: i.e., describing native speakers' particular Gs and limning the properties of FL that allows humans to acquire specific Gs) are clearly projects within the larger domain of cognitive psychology. The third project, MP, is a species of speculative evolutionary biology. What unites the three projects is the aim to uncover the fundamental properties of FL that undergird human linguistic facility.

There is a clear methodological route from the first project to the third. It is idle to speculate about the properties of a meta-capacity that allows for the acquisition of specific Gs before one knows anything concerning the properties that specific Gs have (NOTE: e.g., what kinds of rules do these Gs contain? What kinds of entities do these rules manipulate? What kinds of domains are these rules restricted to?). Once one has some descriptions of individual Gs, it becomes fruitful to ask *why* natural language Gs have the properties they have. It also becomes possible to use these Gs as probes into the human meta-capacity to acquire such individual Gs. An adequate theory of FL will allow for the acquisition of *these kinds of Gs and not others*. So, we will look for theories of FL that constrain the G acquisition process so that they output human kinds of Gs and do not output ones that we do not find in humans.

Once we have an outline of (some) properties of FL that are serviceable we can fruitfully ask yet another follow up question: why do humans have FLs with the properties they do have and don't have properties that they don't have. Again, without some idea of what these general FL properties are (or could be) asking this question is sterile. It is in this sense that MP had to follow investigations into the properties of FL and that investigation of the properties of FL had to follow inquiry into the properties of language specific Gs.

If one's interest is in the structure of FL, this is the only way for an empirical inquiry to fruitfully proceed. And that is (more or less) the way that the Generative Enterprise has unfolded, leading to the current MP questions.

Now, as Chomsky has rightly insisted, programs (like MP) are not true or false, they are fecund or sterile. Sterile programs set a research agenda that is premature. Fecund ones set agendas that are timely. Like many things, there is a tide in the affairs of research programs and, as with the affairs of men, timing is everything. So, a question worth asking is whether after about 25 years the MP the program has proven to be fecund or sterile. In my modest opinion, it has proven to be wildly successful. How so? Because MP has birthed *theories* that have proven to be deeply explanatory, and plausibly, even true. In fact, here is a useful methodological indicator of fruitful programs: they generate interesting theories with non-trivial empirical support. Get enough interesting explanation and enough non-trivial empirical support and you have a fair claim to being (at least roughly) true. In what follows, I will discuss the main minimalist *theory* within MP, something I dub the Merge Hypothesis (MH). I will then suggest that we extend this hypothesis by adding to it a general principle that I dub the Fundamental Principle of Grammar (FPG). (NOTE: I was tempted to call this principle the Central Dogma. This is the tongue in cheek principle that guided a good deal of work in molecular biology over the last fifty years. I refrained for two reasons. First, linguists are by and large humorless and so the joke would fail and lead to nothing but confusion and recrimination. Second, linguistics is sadly not as advanced a field as molecular biology so stealing the moniker would appear chutzpadik. However, the intended role of the FPG is intended to be like that of the Central Dogma; it is a proposed principle of grammatical organization that restricts how FL operates. Moreover, like the Central Dogma, it is likely too strong. The hope is that the FPG can play the same constructive role within MP investigations that the Central Dogma played within the development of molecular biology.) I then show off some nice properties of this Extended Merge Hypothesis (EMH) and call it a day. After all, that is about all that one can do in fifteen pages. This will, of course, serve as a longish advertisement to a larger project that is currently under construction. Let's begin.

1. As mentioned, MP is an exercise in speculative evolutionary biology. We explain why FL has the properties it has by explaining how it arose in humans, the "it" being that which has these properties.

As Chomsky has noted, a few general empirical facts condition any adequate account of the emergence of FL. The first is that FL has arisen relatively recently in the species. It became a species-specific capacity roughly 100,000 years ago. Second, since its arrival on the biological scene it appears to have remained stable with respect to its fundamental properties. Both these facts suggest that FL arose in humans as the result of a small number (preferably one) of simple adventitious mutations which, in combination with the cognitive and computational properties of our pre-linguistic ancestors together sufficed to produce a system like our FL. The main MP conceptual conceit, then, is that FL endowed minds are *minimally* different from those of our ape precursors and that this minimal difference is all the secret sauce required for the human brain to secrete an FL with properties like ours. So, an important step in the MP project is to propose and explore the properties of plausible simple computational candidates to fill the role of cognitive newcomer.

There is a third boundary condition on any adequate account; the properties that sixty years of Generative research has discovered FL to have. More specifically, *Government-Binding Theory* (GB) consists of a bunch of generalizations that describe how natural language Gs are structured. Let's (tendentiously) call these GB generalizations the *laws of grammar*. Here are some example laws:

(1) a. Gs generate an unbounded number of distinct structures that pair a specific meaning with a specific articulation.

　　b. Phrases must be headed.

　　c. Anaphors must be bound in their domains.

　　d. Pronouns cannot be bound in their domains.

e. Nominal expressions must be case marked.

f. Traces must be licensed.

g. Arguments must be θ-marked.

h. PRO can only appear in ungoverned positions at S-structure.

i. And many, many other "laws" of the same type.

GB *theory* offered explanations for these laws, some deeper than others. However, what is relevant here, is that GB offered a largish menu of such laws which were (relatively) empirically robust cross linguistically. Why is this important? For the reason mooted above: we can use these laws as targets of explanation as we have good reason for thinking that they reflect basic architectural features of human FL. As what MP aims to do is *explain* why human FLs have the properties they do have by considering how FL could have arisen, it greatly facilitates investigation knowing what *kind* of FL MP is looking for. Actually, let me say this more strongly. The whole MP project of considering the emergence of FL in the species only really makes scientific sense once we know what kind of thing FL is (i.e., what properties FL has). And this awaited theories that described and explained these properties, theories like GB.

So, given this we can now frame the MP project: find a simple change to the mental economy of our pre-linguistic ancestors which when combined with the rest of their mental inventory of operations and principles yields a cognitive capacity with (more or less) the properties that GB ascribes to FL. That's the project. What makes MP so interesting, to my mind, is that it has spawned a hypothesis as to what the cognitive novelty is. The theory is the Merge Hypothesis (MH) and the conjecture is that Merge, when added to our pre-linguistic cognitive and computational capacities, suffices to derive (more or less) the laws of GB. (NOTE: I keep saying "more or less" because as in any such theoretical exercise the deeper theory will not preserve *all* the features of the target theory. This is fine. Of course, we would be happy to accept a theory that goes the whole nine yards and derives the whole iguana. However, we are realistic and know that every attempt will fall short in some respects. What to make of this, is, of course, a very knotty methodological question. Happily, I intend to completely ignore it here.)

2. MH comes in various flavors. All versions of MP adopt a weak conception:

(2) The (Weak) Merge Hypothesis (MH): Merge is a central G operation.

Virtually all MP accounts suppose that Merge is the operation that explains what Chomsky calls "the central property" of language, namely that natural language grammars are able to generate an unbounded number of distinct hierarchically structured linguistic objects that pair a meaning with an articulation. Merge is a combination operation that is part of a recursive specification for an unbounded domain of such objects. Merge is thereby critical to explaining the capacity that humans have to acquire and use grammars with the central property. In short, virtually all MP theories explain the fact that natural language Gs generate unboundedly many hierarchically structured Gs by taking Merge to be the recursive operation that forms linguistic structures from a finite inventory of basic atoms.

However, the (IMO) best Minimalist accounts prize Merge because it can explain properties of FL beyond the central property. So, for example, some Merge based conceptions of grammar can also explain why grammars that have unbounded structure building also have "movement" operations, and why they display reconstruction effects, and why grammar rules are structure dependent, etc. (I return to showing a little of this anon. For a review of some of the additional properties of grammar that Merge explains see Hornstein (2017) [4].) Building on this I want to suggest that we investigate an extremely strong version of the Merge Hypothesis, one that incorporates what I will dub the "Fundamental Principle of Grammar" (FPG).

(3) The Fundamental Principle of Grammar (FPG): α and β can be grammatically related.

(G-related) only if α and β have merged (If we adopt some conception of Greed, then we can strengthen (3) to a biconditional. Thus, if every G operation "checked" some feature and we understand feature checking as proxy for establishing a G dependency, then (3) can be replaced with (3′): (3′) α and β can be grammatically related (G-related) if and only if α

and β have merged; To my mind, this makes a Greed like principle theoretically attractive, despite its empirical challenges.)

(4) The Strong Merge Hypothesis (SMH): *All* grammatical dependencies are mediated by Merge.

The FPG and SMH are effectively different ways of saying the same thing. The FPG says that expressions can only interact grammatically if they Merge. The SMH says that Merge establishes *all* the grammatical dependencies. (3) and (4) gain considerable bite if we take GB to have offered an inventory of FL grounded grammatical dependencies with their associated properties. The SMH then is the conjecture that the SMH/FPG can deduce/explain all the GB identified Laws of Grammar. (NOTE: The careful reader will have noted that *if* the aim is to derive the GB laws of grammar, then the project takes these laws as (more or less) TRUE. In other words, in what follows I am going to assume that the generalizations discussed are accurate and will not consider any counter-evidence regarding their accuracy. This should not be interpreted as saying that whether they are accurate is unimportant. Rather, it is not that important for evaluating the coherence and interest of the proposed project. I mention this because some reviewers, in my opinion, got lost in the empirical thickets and missed the point of the paper.)

The reader no doubt appreciates how far reaching a proposal the SMH is (indeed, I would be surprised if the main reaction is that the SMH is less *far* reaching than *over* reaching). Here are some immediate consequences: if α binds β then α and β must have merged. If α case marks/θ-marks β then α and β must have merged. If α controls β then α and β must have merged. If . . . you get the idea. So, given the SMH not only is the Central Property an explicandum of the Merge Hypothesis, so too are the basic features of the GB Binding Theory, Control Theory, Case Theory, etc. Indeed, the SMH incorporating the FPG *implies* a Movement Theory of Binding and A Movement Theory of Control as well as a movement approach to Case, Agreement, and any other dependency that GB has proposed as being in the purview of grammar.

Observe that in one important sense, SMH points to a fundamental mistake in the GB conception of FL. GB is a modular theory in that its subparts are informationally distinct components. What I mean by this is that a core feature of the GB theory of FL is its claim that Gs are modularly organized. In particular, the primitives, operations and conditions of the various sub-parts of GB involve different basic predicates, relations and locality domains. Thus, for example, the GB theory of the base trucks in notions like θ-roles and heads that assign such roles under the relation of government. Case theory also assigns features to nominal dependents but these differ from θ-features, are assigned by different heads and in configurations different from those where θ-roles are assigned. Binding Theory differs yet again. It divides the domain of nominal expressions into anaphors, pronominals and R-expressions and requires that the former be locally bound, the second be locally free and the last be free everywhere. It exploits notions like c-command, domain, and indexing. And so on, and so on and so on for the other sub-modules of the grammar. There are strands that unify these sub-domains of the grammar, but within GB each module is (largely) informationally distinct from the others in identifying different relations between different expressions in different grammatical configurations. In other words, though descriptively empirically robust, from the perspective of the SMH, GB is overly complex in being overly modular.

So, if the SMH is on the right track then the internal structure of FL is very non-modular in that all the apparently different relations GB lovingly describes are actually all instances of the same basic Merge operation (NOTE: albeit driven perhaps by the necessity to check different "features"). In this the logic of the SMH *program* of research is the same as that which animates Chomsky's (1977) [5] investigations of movement and islands. Just as Chomsky (1977) aims to eliminate constructions by unifying them in terms of a general movement operation 'Move α' plus the checking of idiosyncratic criterial features, the SMH project aims to eliminate the different GB modules by unifying all the specified dependencies in terms of Merge plus the checking of distinct criterial features. So, just

as constructions resolve into a general movement part and a construction specific set of features so modules resolve into Merge established dependencies and the checking of some specific features (e.g., case, Wh, Agr, θ-roles, etc.).

Before illustrating that this might not be as hopeless a project as one might think, let me locate the SMH by contrasting it with two other grammatical conceptions.

First, as the details illustrate below, the FPG relies on the classical conception of the constituent; a unit with a label. Simple constituents are atoms. Complex constituents are combinations of sub-constituents that are labelled. This fits well with the earliest minimalist conception of Merge, but fits poorly with some contemporary ones where labeling is divorced from Merge. In what follows I adopt the older conception for that is the only one that allows a workable version of the FPG given the simplicity of Merge's combinatorics. What I mean here should become clearer below. For the nonce, I want to emphasize that FPG and the SMH take the classical constituent to be the fundamental locus of grammatical action. Everything of grammatical importance takes place within constituents. I sort of like this feature of the SMH a lot.

Second one should not confuse the SMH with the Strong Minimalist Thesis (SMT). The latter is a conjecture about how Gs optimally mediate between the non-linguistic interfaces. The SMT identifies two, the articulatory and semantic interfaces and conjectures that Merge is the simplest operation for linking the two while at the same time addressing the Central Property.

The SMH is far more specific. It holds that Merge is the only available operation for establishing G dependencies *as identified by GB*. (NOTE: I am speaking more categorically than I should here. The SMH is compatible with the idea that in addition to Merge there is a book-keeping operation that sees to it that features get "checked" through the course of a derivation. Thus, feature checking may exist, though I doubt that it is a specifically linguistic operation. Cognition in general involves feature checking. What is special about linguistic feature checking if the SMH is correct is that it (i) takes place very locally within constituents and (ii) it involves linguistic (rather than say, visual or haptic or auditory) features. (i) is a consequence of the FPG and proscribes operations that allow for "long distance" checking as exists, for example, in grammars that incorporate a Probe-Goal architecture. (ii), or the features alluded to, is a problem for MP at the moment. Where do these features, particularly the syntactic ones, come from? In the best of all worlds, they need not be seeded withing FL but can be acquired using general learning procedures given the right data sets. However, how exactly this happens is not understood at present. Like all well-educated Minimalists, I will therefore leave the question unaddressed in what follows.) The SMH is far more specific than the SMT and can be true even if Merge is not the optimal operation for linking the interfaces. Of course, the two theses are also compatible. I could be that Merge is the secret sauce that gave us FL and that it is the only G operation that is available to code grammatical dependencies. That said, the conceptions are different and should not be confused.

3. The SMH incorporates a version of the MH and so inherits its explanatory virtues. Let's consider some.

First, we need to technically specify the Merge operation (see (6)) and the inductive definition that embeds it (see (7)).

(5) a. If $\alpha$ is a lexical item then $\alpha$ is a Syntactic Object (SO).

     b. If $\alpha$ is an SO and $\beta$ is an SO the Merge($\alpha,\beta$) is an SO.

(6) For $\alpha$, $\beta$, SOs, Merge($\alpha,\beta$)$\rightarrow \{_{\alpha/\beta}\ \alpha,\beta\}$.

The inductive definition in (5) specifies an infinite number of SOs in terms of Merge. The specification of Merge in (6) has two components. One part takes inputs and groups/combines them. A second part categorizes/types the resulting grouping by having one of the inputs to the combination label the complex. The current favorite definition of Merge leaves out the second step, implicitly assuming that categorization is not required for syntactic operations. The definition in (6), in contrast, adopts the early Minimalist conception, which enshrines the constituent as the keystone syntactic structure.

I assume, following standard practice that the combination operation Merge incorporates is very, very simple. What's this mean? It means that the combination operation does *nothing but* combine the input elements. "Nothing but" means it does not alter the inputs in any way, preserves whatever information they contain in the output, and *merely* groups them (e.g., does not linearly order them for example). From this it follows that Merge incorporates both sub-parts of the No Tampering Condition (NTC: Inclusiveness and Extension) (NOTE: Observe, the NTC is a consequence of the simplicity of the combination operation, not an explicit separate condition that regulates it.).

The label operation that types/categorizes the combinations is also conceived to be very simple. The category of the combination is identified with the category of one of the inputs. (NOTE: Chomsky 1995 [6] has a nice discussion of how this is a simple procedure. Which input labels the output, I assume following Chomsky 1995, is not a matter of syntax. Why only *one* input serves as label is also addressed, but not, to my mind, convincingly. I discuss this question in Hornstein 2009 [7]. The gist of the account is that labeling serves to close the combination operation in the domain of the lexical items thereby creating an equivalence class of expressions with the lexical atoms as moduli. This means that a complex *must* be mapped to *one* of the atoms thereby prohibiting both inputs from providing labels.) In other words, labeling is endocentric, reflecting the core idea behind GB's X'-theory. All in all, then, the Merge operation adopted in (6) returns to the earlier Minimalist conception of the operation.

As noted, the specification of Merge in (6) differs from the operation assumed in much contemporary Minimalist theory. Specifically, it is more complex than the current dominant version in that it incorporates both a combination operation (wherein two SOs are grouped to form a set) and a labelling operation (wherein the combination is typed by the label of one of the combining elements). Plausibly, this a step backwards theoretically as Minimalists are after the *simplest* possible Merge operation and clearly adding labeling complexifies the operation. This claim requires consideration but I will not give it any here. Why not? The labeling (6), we shall see, is required to technically state the FPG while respecting the NTC. As this paper is in service of showing off the virtues of the EMH which incorporates the FPG I will assume what is required to make this train run on time (NOTE: However, just to make it clear that I am not (merely) running away from the issue and that I have some response to this reasonable point, here are three quick observations. First, the current theory does not eliminate labeling, rather it relocates it and makes it part of the Spell Out operation. Thus, complexes are typed via the labeling algorithm on their way to (at least) the CI interface. So, labeling is not eliminated from the grammar, it is just relocated and so it is not clear that the *overall* system is any simpler than the present one. Second, whatever its other virtues, the labeling algorithm is quite a bit more complex than free endocentric labeling. In fact, the real issue is not whether we need labeling but what kind we need (endocentric or labeling algorithm) and where it applies (as part of the syntax or on route to interfaces). Third, the EMH/FPG allows for the entire elimination of the Probe-Goal/Agree rule architecture. This is *required* without labels, as I will make clear below. It is entirely eliminable with them. So again, the overall grammar without labels requires at least the addition of a the Agree/Probe-Goal system. There is more, but this should suffice here.).

Ok, let's show off. First, as the EMH incorporates the MH it can claim all of its explanatory victories. So, for example, just as the MH serves to generate an unbounded number of hierarchically organized objects, so too does the EMH incorporating Merge in (6). Here is an illustration to "prove" this. (NOTE: This example leaves out a few steps for the incorporation of T(ense) and other functional material into the generated structure, but you, dear reader, can no doubt fill in the requisite details.)

(7) The king of Georgia hugged the portrait of Lenin.

(8) a. As *of* is an SO and *Lenin* is an SO (5b)/(6) licenses the construction of the SO

{$_P$ of, Lenin}. (NOTE: I designated the head as 'P' rather than 'of'. The latter is more correct but I will use standard assumptions concerning the syntactic features of the head

and use them to label the expression. This is mainly to save typing effort. The "real" head is 'of' here and the relevant lexical item in all that follows.)

b. The SO {$_P$ of, Lenin} merges with the SO *portrait* to form the SO {$_N$ portrait, {$_P$ of, Lenin}}.

c. The SO *the* merges with the SO {$_N$ portrait, {$_P$ of, Lenin}} to yield the SO {$_N$ the,{$_N$ portrait,{$_P$ of, Lenin}}}. (NOTE: I am here tacitly rejecting the DP hypothesis and assuming that 'portrait' is the head of the nominal expression. Nothing hangs on this here, but see below in the discussion of selection.)

d. The SO *hugged* merges with the SO {$_N$ the,{$_N$ portrait,{$_P$ of, Lenin}}} to yield the SO {$_V$ hugged, {$_N$ the,{$_N$ portrait,{$_P$ of, Lenin}}}}.

e. The SO *of* and the SO *Georgia* merge to form {$_P$ of, Georgia}.

f. The SO *king* and the SO {$_P$ of, Georgia} merge to form {$_N$ king, {$_P$ of, Georgia}}.

g. The SO *the* and the SO {$_N$ king, {$_P$ of, Georgia}} merge to form the SO {the, {$_N$ king, {$_P$ of, Georgia}}}.

h. The SO {$_N$ the, {$_N$ king, {$_P$ of, Georgia}}} and the SO {$_V$ hugged, {$_N$ the, {$_N$ portrait,{$_P$ of, Lenin}}}}merge to form the SO {$_V${$_N$ the, {$_N$ king, {$_P$ of, Georgia}}},{$_V$ hugged, {$_N$ the,{$_N$ portrait,{$_P$ of, Lenin}}}}}.

Second, just like MH, the EMH unifies structure building (E-merge) and Movement (I-merge). Again let's illustrate, this time giving a more abstract example. Consider the derivation of (9b) form (9a) in (10):

(9) a. {$_γ$ γ, {$_β$ λ, {$_α$ α, β}}}.

b. {$_γ$ β, {$_γ$ γ, {$_α$ λ, {$_α$ α, β}}}}.

(10) The SO {$_γ$ γ, {$_β$ λ, {$_α$ α, β}}} and the SO β (within {$_γ$ γ, {$_β$ λ, {$_α$ α, β}}} merge to form

{$_γ$ β, {$_γ$ γ, {$_α$ λ, {$_α$ α, β}}}}.

Note, that just as in MH, (9b) codes the movement of β by having (an occurrence of) β occur in two different sets (one headed by α and one headed by γ). Importantly the same Merge operation applies here as applied in the derivation of (8). Hence, the unification of structure building and movement.

And like the MH, the EMH derives the c-command condition on movement. More exactly, as movement is a species of Merge (viz. I-merge) and Merge incorporates the NTC, movement must target c-commanding positions. Again, consider a simple illustration in (11).

(11) a. {{α,β},{γ,δ}}.

b. {{{γ,α},β}, {γ,δ}}.

c. {{γ,{α,β}},{γ,δ}}.

d. {γ,{{α,β},{γ,δ}}}.

Both MH and EMH incorporate the NTC. The NTC prohibits deriving structure (11b) from (11a). Here, we Merge γ with α. The output of this instance of Merge obliterates the fact that {α,β} had been a unit/constituent in (11a), the input to Merge. EC prohibits this. If EC is understood as the requirement that Merge leave undisturbed the structure of the input, then (11b) is not a licit instance of I-Merge (note that {α,β} is not a unit in the output. Nor is (11c) (note that {{α,β},{γ,δ}} is not a unit in the output). Only (11d) is grammatically kosher for all the inputs to the derivation (i.e., γ and {{α,β},{γ,δ}}) are also units in the output of the derivation. A new relation has been added, but no previous ones have been destroyed. Note that a corollary of the NTC is that Merge operations must always take place at the root, and so we derive that a Merge based derivational system that analyzes movement as I-merge derives the fact that movement targets c-commanding positions. (NOTE: To be more precise, this only holds for standard single rooted derivations. A reviewer rightly pointed out that this line of reasoning originates with Epstein (1999) [8]. It shows how given a conception of derivations using a Merge like understanding of concatenation/structure building one gets a relation like c-command as a by-product. The above closely conforms to Epstein's (1999) understanding of the issues.)

4. There are other properties of FL that MH derives that EMH similarly derives. However, let's now turn to some cases where the labeling part of Merge does some explanatory work.

Let's start with θ-marking. The FPG says that the only way for one expression to θ-mark another is if the two Merge. There are two rough configurations for θ-marking within GB. The internal argument (IA) is θ-marked under sisterhood with the predicative head (P) it merges with, as in (12a) and the external argument (EA) is θ-marked by merging as the specifier of predicate (P) that θ-marks it, as in (12b).

(12) a. {P, IA}.
    b. {EA, {P ... }}.

The FPG can unproblematically apply to (12a) as P and IA are sisters here. Not so (12b). More specifically, *if* θ-marking is a relation between a predicate/head P and an argument/XP then the FPG requires that P and XP merge for θ-marking to take place. (12a) allows for this without labels. (12b) does not. Why not? Because EA does not merge with P in (12b). Moreover, if Merge subsumes the NTC, EA *cannot* merge with P in (12b)! So given the FPG either internal and external arguments are fundamentally different kinds of θ-dependents or EAs must be able to Merge with Ps in configurations like (12b) in some extended sense. (NOTE: *any* non-complement θ-role presents the same problem as IAs. So, if there are three arguments (e.g., with ditransitives) then only one can be the immediate sister of the head. The assignment of the other θ-roles is problematic without labels if we assume such roles are assigned via heads.) Labels support the requisite extended sense. Here's how. If we allow that labels exist, then just as IAs merge with a label of P in (12a) EAs can merge with a label of P in (12b). We represent this option in (13), where the constituent containing P as head carries the label 'P.' Labeling is then a way for expressions that have merged once to merge again without violating the NTC. They do this by labeling the complexes that contains them.

(13) {EA, {$_P$ P ... }}.

None of this is original. It simply regurgitates the assumptions about endocentricity, projection and labeling that is part of standard X'-theory and reframed in terms of labels as in early Minimalist theory. What is relevant here, is that a labeling convention of this kind is *needed* to make the FPG operational given the simple conception of Merge that embodies the NTC. Without labels specifiers cannot grammatically "talk to" heads under Merge. Once we add labels, so discoursing is trivial. If we adopt the FPG as *the* regulative, fundamental, ur-principle of grammar, then an operation like endocentric labeling must exist. One might be tempted to make the same point as follows (snicker! snicker!): *given the FPG, labeling is virtually conceptually necessary.*

We can go a bit further, and so I will. *If* external arguments are marked by predicative lexical heads or functional heads like *v*, then the FPG requires that external arguments be θ-marked *by merging with a projection of that predicative head*. For example, take a transitive verb, if the EA is marked *v*, then EAs must merge (with a label of) *v* to be so marked. Additionally, this implies that EAs must begin their derivational lives as sisters to a projection of *v* (e.g., as in (14)). To repeat, this follows directly from the EMH/FPG and the assumption that EAs are θ-marked by *v*s. Thus, the Predicate Internal Subject Hypothesis (i.e., the fact that subjects are base generated within the verbal projection) is a corollary of the EMH/FPG.

(14) [$_v$ EA [$_v$ v [$_V$ V IA]]].

Let me emphasize this point by noting a less general alternative. We can *stipulate* that θ-marking only takes place under Merge and arrive at the same result. However, if we have other kinds of grammatical operations (e.g., Agree/Probe-Goal, Binding), then the fact that Gs require their subjects to be generated predicate internally *does not yet follow*. It only follows if we can explain *why* θ-roles cannot be assigned under Agree/Probe-Goal or Binding. Were this an option then the EA need not be generated as sister to the vP. (NOTE: Indeed, such a mechanism was suggested for control structures in Manzini and Roussou 2000 [9] and could easily be extended to all cases of EA θ-role assignment.) It is precisely by blocking this theoretical option that the EMH/FPG explains why EAs must

be base generated as sister (of a projection) of the θ-assigning predicate. In other words, only by requiring that *all* grammatical dependencies be Merge mediated do we derive the fact that subjects must be base generated as low as they appear to be generated (i.e., predicate internally). And this is why the EMH/FPG *explains* why Gs have predicate internal subjects.

5. Let's move onto selection/subcategorization (henceforth S/S) and its idiosyncrasies. S/S is a head-to-head relation that occurs under sisterhood. More specifically, S/S is restricted as follows:

(15) A head X can only select/subcategorize the head Y of its sister.

Thus, it can only occur in configurations like (16) where Y is the head of the (possibly complex) sister of X.

(16) {X, {$_Y$ ... Y ... }}.

Examples of selection are cross linguistically robust and, from what we can tell, largely idiosyncratic. So, for example, we say *interested in* not \**interested on/from/with* and *angry at/with* but not \**angry from/against/to*. Ditto with an *interest in* not \**on/out* and my anger *at/with* but not \**from/against*. Nor need the selection restrictions generalize across categories as in the examples above. As Merchant has noted we say *proud of/\*in* but *pride in/\*of*. There seems to be little rhyme or reason behind these restrictions.

Moreover, they extend to functional categories as well. So, *for* complementizers select for infinitival T-heads while finite *that* selects for finite T-heads. All of this is well known.

(17) a. John prefers for/(\*that) Mary to leave.

b. John said that/(\*for) Mary left.

What is relevant here is that this, again, follows seamlessly from the EMH which incorporates the FPG. As selection is a grammatical dependency it must be licensed under Merge. This means that it is only in configurations like (16) that selection *can* apply given the FPG.

However, there is more. (15) states that selection is *limited* to heads. Call this the "Periscope Property" (so dubbed in Hornstein, Nunes, and Grohmann (2005) [10]). The Periscope Property highlights three facts about selection: (i) that in a configuration like (16) X *can* select for Y (ii) that X *cannot* select for anything thing else within Y (i.e., a head X cannot police elements in the specifier or complement positions of Y in (16) even though by many measures it is as close (and maybe closer) to these than it is to the head), and (iii) that the selection relation is *linearly* unbounded (i.e., there can be as much stuff as you like separating X and Y and X can still work its magic on Y). So, for example, consider the structure underlying (17a). It is roughly that in (18):

(18) John prefers (for ... to leave).

Note that an arbitrarily big nominal phrase can occupy the subject position without interfering with selection of *to* by *for* (e.g., insert *Seven of the eight large unicorns that ran through the living room* into ' ... '). Nor will an intruding simplex subject (e.g., *it*) block access to *to*. So, selection is a relation between a head and the head of its sister and only the head of its sister.

Note that all of this follows directly from the FPG. The FPG requires that selection be mediated by Merge. Merge generates structures like (16) by merging an X labeled expression with a Y labeled expression. In other words, it merges X and Y. If selection is governed by Merge then only X and Y can grammatically interact. Why can X not select a complement or specifier of Y? Because it does not merge with these (indeed, given the NTC, it cannot merge with these). The only expression it merges with is labeled Y so its grammatical business is restricted to Y and Y alone. Voila, we derive (15)!

Two more points and we move on. First, the account relies on Merge being the *only* operation available for forming grammatical dependencies/structures. To see this, consider how this explanation *fails* if we assume that in addition to Merge grammars allow operations like Agree/Probe-Goal to from grammatical dependencies. In such a grammatical world a head could agree with another head that it probed under c-command (the standard assumption in much of the Minimalist literature). Now take a look at the

configuration in (16) once again. Were it possible to select under Agree/Probe-Goal it is quite unclear why specifiers (and adjuncts) are immune to S/S. In (16), X c-commands everything within the projection of Y including any specifiers, complements, or adjuncts contained within it. So were Agree/Probe-Goal a potential licit grammatical operation then one would expect these positions to be selectable/subcategorizable by higher heads. Indeed, heads do typically probe far down into a phrasal sister to license, for example, WH movement. But, as noted, FL/UG does not allow this species of selection, which would be somewhat surprising if selection could be executed under Agree/Probe-Goal. In contrast, the noted restrictions follow directly if one adopts the FPG, but *only* if one restricts the stable of admissible grammatical operations to Merge alone and bars potentially long-distance operations like Agree/Probe-Goal.

Let me be clear. It is *possible* to stipulate that selection is restricted to hold exclusively between heads even assuming Agree/Probe-Goal is a licit grammatical operation. It is even possible to propose that a conspiracy of assumptions prevents Agree/Probe-Goal from selecting specifiers and complements (e.g., if we assume that specifiers can never be simple heads it might be possible to prevent selection via some notion of economy wherein the selected head is always more proximate to the selecting head than anything else within the phrase could be. (NOTE: Though even this assumption would not suffice. We would need to supplement it to evade the "relativized" part so that selection could not "ignore" non featurally relevant interveners. Nonetheless, technically, I have no doubt this could be done (see, e.g., Collins 2002) [11].)) However, ancillary assumptions are required (and, in my opinion, are not particularly well-grounded). So, things can be made to fit but explanatory power is (to my mind, quite clearly) sacrificed.

Second, the explanation of the Periscope Property offered here requires that Merge combine labeled expressions. Without them, in (16) X cannot get hold of Y via Merge as merging X directly to the head Y would violate the No Tampering Condition. In fact, the only licit way of X selecting Y in structures without labels would restrict it to cases like (19) in which X and Y are lexical atoms. This is a coherent possibility. It *could* have been the case selection only occurred when lexical atoms merged. However, this is false. Though selection is structurally restricted it is not restricted to bare heads. (NOTE: The EMH/FPG has other consequences that I do not discuss here. To take one example, it implies, given conventional assumptions about the class of caser assigners, that case assignment will impact potential scope readings. As is well known, Lasnik and Saito 1991 [12] demonstrated this interdependency between scope value and scope. Just as interesting, in my view, the FPG fits poorly with certain more contemporary conceptions of case, in particular Dependent Case Theory. Again, I will not discuss any of this here but will discuss it in a forthcoming development of the basic ideas outlined here.)

(19) {X,Y}.

6. Let us now turn to construal. Construal relations are non-local anaphoric dependencies. They come in three flavors: Control, Reflexivization/A-effects, and Pronominalization/ B-effects (Reflexivization includes reciprocals as well and may extend to other kinds of anaphoric dependencies as well. In GB, for example, A-traces were treated as kinds of silent reflexive/A-anaphors.). Characteristic of all three is that there is an antecedent that "binds" an anaphoric dependent thereby contributing to the dependent's interpretation. For example, (20a/b) are interpreted so that John is interpreted as having the semantic value of the external argument of *expect* and the external argument of *win*. (20c) allows for such a reading but does not require it. Thus, construal involves interpretively relating expressions to *multiple* θ-roles.

(20) a. John expects PRO to win.

　　b. John expects himself to win.

　　c. John expects that he will win.

Within GB this dependency was technically executed via co-indexation subject to various locality conditions. Importantly, GB assumptions *required* treating construal depen-

dencies as distinct from movement dependencies. Why? Because given how D-structure is defined within GB, movement into θ-positions is illicit.

It is worth pausing on this point. *Were* movement into θ-positions allowed, movement suffices to provide an expression with multiple θ-dependencies. In fact, we could simply say that an expression acquires θ-roles by merging into θ-positions. We already say this for E-merge. Indeed, all Minimalist accounts allow an expression to acquire a θ-role by E-merging into a θ-position. And, *if* (as all Minimalist accounts assume) E and I Merge are the same operation then there should be no principled reason why I-merging into a θ-position should be less able than E-merge to pair an expression with a θ-role. *And* once D-structure is eliminated as a level (again something that most Minimalist accounts do) then there is no principled grammatical reason why it is impossible to E-merge into a θ position and then I-merge into another (and another, and another). In other words, once phrase building (E-merge) and movement (I-merge) are unified as different faces of Merge and once D-structure is removed as a syntactic level it becomes possible to treat construal as movement. And, let's drop the obvious second shoe; if one assumes that Merge is the sole grammatical operation (i.e., no Probe-Goal/Agree, no co-indexation) then the only way of grammatically establishing construal dependencies is via serial applications of Merge (both E and I) into multiple θ-positions. In other words, the EMH/FPG force a movement account of construal relations.

Let's phrase this conclusion another way. The EMH/FPG requires treating all construal relations as mediated by (I-)merge and so, parasitic on movement. One way of describing this, is that a corollary of the EMH/FPG is that all construal dependencies "live on" chains. From the perspective of the EMH classical "movement" chains differ from "construal" chains in just one way: in movement chains only one "link" occupies a θ-position, while in construal chains many links do. In other words, both classical movement chains and construal chains are formed via Merge but the number of θ-positions Merge targets differ in the two cases. The simplest version of the EMH should be that in all other respects the two kinds of chains should be the same. (NOTE: I should say "simplest" because the EMH is compatible with there being differences among chains that have nothing to do with their being products of (I-)merge. For example, within GB control structure anaphors are phonetically null, while in reflexive and pronominal binding structures the anaphoric dependents are phonologically overt. If there are principles of grammar that care about this difference, and there is no a priori reason why there shouldn't be, then this may distinguish some properties of chains from others. I will abstract from these concerns here.)

There is evidence supporting this Panglossian possibility. For example, just as Reflexivization obeys the Tensed S Condition, so too does A-movement and Control (c.f. (21)) (Ditto for the Specified Subject Condition but I don't have room to illustrate this here.):

(21) a. $Mary_1$ believed $herself_1$ to have won.

    b. *$Mary_1$ believed $herself_1$ had won.

    c. $Mary_1$ was believed $t_1$ to have won.

    d. *$Mary_1$ was believed $t_1$ had won.

    e. $Mary_1$ expects $PRO_1$ to win.

    f. *$Mary_1$ expects $PRO_1$ will win.

Similarly, just as (A-)movement/Merge must target a c-commanding position so too the antecedent of a reflexive/PRO must be in a c-commanding position. Indeed, as I have argued (repeatedly and *ad nauseum*) in other work, we can derive a good chunk of the properties of reflexive and control structures by simple assuming that the antecedent is the head of a chain containing the reflexive/PRO as a link. And this "movement" analysis of these construal analyses follows from the FPG. Thus, the EMH/FPG explain the basic properties witnessed in construal dependencies.

We can go a bit further. If all construal dependencies are chain dependencies, then, given what we know about the typology of chains we expect to find two kinds of construal dependencies; those that piggy back on A-chains and those that piggy back on A′-chains. (NOTE: The reviewer made a comment that I agree with: "I have to admit that any mention

of the A/A′ distinction makes me squirm, it is an ugly remnant of the GB days (NOTE: I would not even know how to define it any more, certainly not as an inherent property of particular positions)." I applaud the squirming and loath that the distinction is still required, even if poorly understood. That it is required seems clear as the there is little doubt that what we classify as A-chains have very different properties from those we classify as A′ (span of the dependencies, properties of the links, WCO effects, SCO effects, binding effects). So, an A/A′ distinction seems indispensable, even though as the reviewer rightly observes, it is currently theoretically opaque. I do have some thoughts of possible ways of grounding the distinction in more fundamental concepts, but this is not the place to do so. That said let me oracularly pronounce: say phases exist, we might be able to distinguish intra-phasal movement from inter-phasal movement and use this difference to mimic the A/A′ difference. Maybe.) As noted, reflexive and control chains are plausible examples of the former. Pronominal binding (as in (20c above) plausibly piggy backs on the latter. (NOTE: This was first proposed by Kayne 2002 [13]. The FPG requires that some version of Kayne's approach to pronoun binding be correct. See also Ginzburg and Fong (2010) [14].) Here is a quick illustration of how this might proceed.

Consider sentences like (22a). The FPG would analyze them as (22b):

(22) a. Everyone$_1$ said that Mary likes him$_1$.

b. [*Everyone* [said [$_{CP}$ *Everyone* that [Bill [likes *Everyone (= him)*]]]]].

Let's abstract from the process that (morpho-phonologically) converts the lower copy of *everyone* into a pronoun. ((NOTE: I here adopt the proposal in Idsardi and Lidz 1998 [15]. There are other executions, but this one is quite clean and I adopt it here. It should be noted that their approach is reminiscent of the treatment of binding via Reflexivization and Pronominalization rules in Lees and Klima 1963.) [16]). The EMH treats the bound pronouns reading as living on chains with multiple θ-positions (the top and bottom links) and that involve A′-movement via some position (here the embedded Spec CP). (This *is* a case of improper movement, so the EMH requires not only giving up the θ-criterion which prohibits moving into θ-positions, but also requires allowing improper movement.)

Now, the informed reader will know that this proposal faces many challenges. Thus, for example, in addition to permitting movement into θ-positions (as required for the A-chain construal dependencies above) it further requires allowing improper movement as well. (NOTE: More challenging, this analysis suggests that binding should display Island Effects. I have argued that this is indeed the case in some unpublished work. And will lay the fuller argument out in Hornstein forthcoming.) Let's assume that these problems can be finessed. Are there any advantages to treating pronominal binding in this way?

Well, yes, there are. Several. Here is one that I like. If this approach to Pronominalization is correct then we can *derive* the fact that Reflexivization and Pronominalization are (largely) in complementary distribution (or, that Principles A and B apply in complementary fashion). (NOTE: I inserted "largely" to be mealy mouthed at the suggestion of the reviewer. There are some cases where this complementarity is obscured. I don't believe, myself, that these are real problems and think that Reflexivization and pronominalization are actually complementary. See Safir 1997 [17] for empirical arguments to this effect.) The complementarity follows from the observation that reflexives live on A-chains that contain no A′-links and bound pronouns live on (improper) chains that crucially include an A′ link. In sum these two chain types are mutually exclusive structurally and so reflexives and bound pronouns are expected to be in complementary distribution (which they are (ahem, largely)). Furthermore, we expect reflexives to be found more locally than bound pronouns precisely because A-movement steps are "shorter" than A-movement steps are. Again, this seems to be on the right track as Reflexivization into finite complements is prohibited while pronominalization is not. (NOTE: Again, an observation first made in Lees and Klima and incorporated into every subsequent theory of binding.)

It is perhaps worth noting that, to my knowledge, the above inchoate tale is the only explanation on offer as to *why* Principles A and B are complementary. We have known for a very long time that this complementarity is a central feature of binding within natural

language, but we have never had an explanation for why FL imposes complementarity on these two processes. Of course, offering a potential explanation for a long-standing generalization does not make the proffered account correct, but it highlights a potential payoff of thinking along EMH/FPG lines.

7. Nothing on offer above should convince anyone that the EMH/FPG is correct. Frankly, I am somewhat skeptical that it can overcome the many obvious empirical problems with it. Still, it has some very attractive properties given the aim of accounting for the many Laws of Grammar Generative Grammar has uncovered over the last 60 years. The Merge Hypothesis is a very simple theory, with considerable empirical grounding and theoretical power. The EMH takes the logic behind the MH to the limit by elevating the FPG into the central grammatical regulative principle. I have limited myself here to outlining the motivations for it and (hopefully) showing that it is not entirely hopeless. Whether it is more than that, well, only time can tell.

**Funding:** This research received no external funding.

**Institutional Review Board Statement:** Not applicable.

**Informed Consent Statement:** Not applicable.

**Data Availability Statement:** Not applicable.

**Conflicts of Interest:** The authors declare no conflict of interest.

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
