# Peer review of "The Extended Merge Hypothesis and the Fundamental Principle of Grammar"

_philosophies, doi:10.3390/philosophies6040089_

Round 1

Reviewer 1 Report

To be clear, I am not against papers that take a broad perspective and mainly serve as an overview of some topic or field, and that push a particular perspective. However, as I see it, there are two main problems:

  1. For specialists, this paper does not contain anything new, and at various points adequate references are lacking. For non-specialists, the overview seems really hard to comprehend/value.
  1. The tone/style is not as it should be in a scholarly journal. The abstract is completely inappropriate, footnote 2 is, well, bizarre, if I may say so. And the paper contains various formulations that do not belong to a neutral, academic register. (This does not mean that the text should be overly formal.)

I will therefore refrain from a detailed discussion here.

Author Response

Not much to respond to. The reviewer does not like the paper. It is written for specialists. Neophytes will find it incomprehensible. What I find a bit puzzling is the suggestion that this is all old hat. So far as I know, nobody has ever suggested any principle analogous to the SMH/FPG. My worry is that it is obviously too strong. The reviewers worry that it is totally uncontroversial. I hope the reviewer is right.

last point about style. It is unconventional. But I have no desire to change this. If this is a problem, the editors should tell me and I will withdraw the MS. 

Reviewer 2 Report

Follow the submission guidelines. 

The research problem and gaps in the field should be identified. 

The paper contains minor grammatical errors, spelling and space. 

The body of article should be typed/titled ( Abstract, Introduction, Methods, results, discussion, conclusion, recommendations and references). 

These elements are missing in the paper. 

Author Response

I have no idea what the third bullet means. I re-read the paper and liked the flow. If the editors want to suggest headings, feel free.

Reviewer 3 Report

  • Lines 64-66: The author should add citations to support the statement here.
  • Lines 84-85: The author should consider long-distance anaphor and Binding Principle B in Vietnamese before s/he presents (1c/d) as "generalizations".
  • Footnote 2: This footnote is not helpful at all to advance the argumentation in the main text. Relatedly, the author should refrain from expressing personal opinions/thoughts in an academic writing; rather, s/he should focus more on facts, data, observations, and argumentation. This way the author would not have to worry/complain about the limited space of the manuscript.
  • Line 90: (1g) should be deleted since it is redundant.
  • Footnote 4: The author 
  • Lines 129-131: The author should consider the grammatical relation between a displacement of X and the lower copies of X. The copies of a displacement object are clearly grammatically related, but they are not merged.

Author Response

Again, not a fan. I like the style and would prefer not to change it. I like expressing personal opinions in a programmatic paper where I lay out my views. Again, if this is something the editors don’t like, I am happy to withdraw the paper.

Long distance anaphors: I added a note asserting that the GB generalizations were for the purposes of the paper taken as true. LDAs are discussed in a much longer MS. 

Reviewer 4 Report

The paper explores a conceptually very appealing hypothesis that A and B can be grammatically related only if A and B have merged (FPG). The hypothesis is far-reaching, with many strong predictions, some of which are discussed in the paper. That the hypothesis makes very strong predictions all over the place is not a weakness; on the contrary, it makes it very appealing since it considerably constraints the power of the system. The paper is programmatic in nature, as such it could not go into the details of the phenomena under consideration, which is understandable and should not be held against it. At any rate, this is an extremely interesting and thought-provoking work; I recommend that the paper be accepted for publication. I have only some minor comments, leaving it up to the author to decide what to do with them.

Comments:

It might be useful to mention (with a bit of comparison) Epstein’s 1999 (Working Minimalism volume) work on c-command.

In some places, it would be good to add a bit of hedging (e.g. reflexivization and pronominalization being by and large/largely in complementary distribution.

The gist of compositional semantics looks similar to the FPG.

It is interesting that one part of Chomsky’s labeling algorithm is essentially automatic, the way it was in the pre-labeling system: it’s the case where a head and a phrase merge (Boskovic’s 2016 TLR paper in fact argues that in this case labeling must take place immediately, tying it to selection)

I have to admit that any mention of the A/A’ distinction gets me to squirm, it is an ugly remnant of the GB days (I would not even know how to define it any more, certainly not as an inherent property of particular positions).

Author Response

These were useful to me. I included a few extra notes in response, aminly to make it clear why I think Complementarity of reflexivization and pronominalization holds. I also agreed in an extra note that the A/A’ distinction should make linguists queasy. Thx for the comments.